# Stimulation of VTA dopamine inputs to LH upregulates orexin neuronal activity in a DRD2-dependent manner

**Masaya Harada[1], Laia Serratosa Capdevila[1], Maria Wilhelm[1], Denis Burdakov[2,3], Tommaso Patriarchi[1,2]***

[1]Institute of Pharmacology and Toxicology, University of Zürich, Zürich, Switzerland;
[2]Neuroscience Center Zürich, University and ETH Zürich, Zürich, Switzerland;
[3]Department of Health Sciences and Technology, ETH Zürich, Zürich, Switzerland

*For correspondence:
patriarchi@pharma.uzh.ch

**Competing interest:** The authors declare that no competing interests exist.

**Abstract** Dopamine and orexins (hypocretins) play important roles in regulating reward-seeking behaviors. It is known that hypothalamic orexinergic neurons project to dopamine neurons in the ventral tegmental area (VTA), where they can stimulate dopaminergic neuronal activity. Although there are reciprocal connections between dopaminergic and orexinergic systems, whether and how dopamine regulates the activity of orexin neurons is currently not known. Here we implemented an opto-Pavlovian task in which mice learn to associate a sensory cue with optogenetic dopamine neuron stimulation to investigate the relationship between dopamine release and orexin neuron activity in the lateral hypothalamus (LH). We found that dopamine release can be evoked in LH upon optogenetic stimulation of VTA dopamine neurons and is also naturally evoked by cue presentation after opto-Pavlovian learning. Furthermore, orexin neuron activity could also be upregulated by local stimulation of dopaminergic terminals in the LH in a way that is partially dependent on dopamine D2 receptors (DRD2). Our results reveal previously unknown orexinergic coding of reward expectation and unveil an orexin-regulatory axis mediated by local dopamine inputs in the LH.

## eLife assessment

This study presents **valuable** findings that expand our view of dopamine release in different brain regions and show that dopamine release in the lateral hypothalamus is related to the activity of orexin neurons. The evidence supporting the claims of the authors is **solid**, although inclusion of tests that directly assess causality of the noble pathways would have been even more conclusive. The work will be of interest of neuroscientists who study the neural basis of motivation.

## Introduction

Dopamine in the ventral and dorsal striatum shapes reward-related behaviors (*Markowitz et al., 2023*; *de Jong et al., 2022*; *Yang et al., 2018*; *Keiflin and Janak, 2015*; *Tsai et al., 2009*); its dysregulation has been associated with several psychiatric disorders, including addiction (*Lüscher and Janak, 2021*; *Lüscher et al., 2020*; *Pascoli et al., 2023*) and depression (*Nestler and Carlezon, 2006*; *Krishnan et al., 2007*; *Deguchi et al., 2016*). It is known that rewarding stimuli evoke dopamine transients both in the ventral (*Kim et al., 2020*; *Patriarchi et al., 2018*) and dorsal striatum *Howe et al., 2013*, and that the stimulation of dopaminergic neurons (*Harada et al., 2021*; *Pascoli et al., 2018*) or terminals (*Yang et al., 2018*) in the striatum is sufficient to trigger operant or Pavlovian conditioning (*Saunders et al., 2018*), as well as conditioned place preference. Instead, aversive stimuli or omission of expected reward delivery cause a decrease in dopamine in the ventral striatum, resulting in negative

reinforcement learning (*Tan et al., 2012*; *van Zessen et al., 2012*) via D2 receptors (*Iino et al., 2020*; *Lüscher and Pascoli, 2021*).

Although the role of the dopaminergic projections to the striatum or mesolimbic dopamine pathway has been investigated extensively (*Kim et al., 2020*; *Cohen et al., 2012*) – their role in encoding reward prediction errors (RPEs) in particular has been a point of focus (*Kim et al., 2020*; *Schultz et al., 1997*) – the role of dopamine in other brain regions is relatively understudied (*Hasegawa et al., 2022*; *Vander Weele et al., 2018*; *Gyawali et al., 2023*; *Chen et al., 2014*). The lateral hypothalamus (LH) plays a pivotal role in reward-seeking behavior (*Gibson et al., 2018*; *Harris et al., 2005*; *Otis et al., 2019*; *Sharpe et al., 2017*; *James et al., 2019*) and feeding (*O'Connor et al., 2015*; *Jennings et al., 2015*; *Jennings et al., 2013*; *Marino et al., 2020*), and several dopamine receptors are reported to be expressed in the LH (*Yang et al., 2019*). The mechanism through which dopamine modulates neuronal activity in the LH, resulting in the modulation of behaviors, has not been established. To the best of our knowledge, there have been no measurements of dopamine transients in the LH during reward-associated behaviors.

The LH is a heterogeneous structure containing glutamatergic and GABAergic neurons, as well as several neuropeptidergic neurons, such as melanin-concentrating hormone-positive and orexin-positive neurons (*Mickelsen et al., 2019*; *González et al., 2016a*). Like dopamine, orexins (also known as hypocretins) are reported to play a pivotal role in reward-seeking behavior (*Harris et al., 2005*; *Borgland et al., 2006*; *Bubser et al., 2005*). Orexinergic and dopaminergic systems are known to have reciprocal connections with each other, and some orexinergic neurons project to dopaminergic neurons in the ventral tegmental area (VTA), positively modulating their activity (*Thomas et al., 2022*; *Baimel et al., 2017*). While there has been extensive investigation into how dopamine modulates orexinergic neuronal activity ex vivo (i.e., acute brain slices) (*Yamanaka et al., 2006*; *Li and van den Pol, 2005*; *Linehan et al., 2015*; *Linehan et al., 2019*), it remains unclear whether and how dopamine transients modulate orexin neuronal activity in vivo (*Linehan et al., 2019*). Advancements in optical tools, such as optogenetics for manipulating dopamine neurons and genetically encoded dopamine sensors for monitoring dopamine transients, have made it possible to precisely control and observe the dynamics of dopamine in neural systems (*Patriarchi et al., 2018*; *Feng et al., 2019*; *Sun et al., 2020*; *Zhuo et al., 2023*; *Wu et al., 2022*; *Patriarchi et al., 2020*). Here, we implemented an 'opto-Pavlovian task' (*Saunders et al., 2018*), in which mice learn to associate a sensory cue with optogenetic dopamine neuron stimulation. Using this task we measured dopamine transients in the nucleus accumbens (NAc), finding that dopamine activity patterns are consistent with previous reports of RPE-encoding dopaminergic neuron activity (*Cohen et al., 2012*). Using the same paradigm, we found that optical stimulation of dopaminergic neurons in the VTA evokes an increase in extrasynaptic dopamine in the LH, where the delivery of a cue preceding a reward also triggers dopamine transients in a way that is consistent with RPEs (*Schultz et al., 1997*). Furthermore, we investigated the regulation of LH orexinergic neurons by VTA dopaminergic neurons and observed a dopamine transient in the LH and an increase in orexinergic neuronal activity during both predictive cue and the delivery of laser stimulation, indicating that the concentration of extrasynaptic dopamine in the LH and orexinergic neuronal activity are positively correlated. Finally, by stimulating dopaminergic terminals in the LH combined with pharmacological intervention, we found that dopamine in the LH positively modulates orexinergic neurons via the type 2 dopamine receptor (D2).

Overall, our study sheds light on the meso-hypothalamic dopaminergic pathway and its impact on orexinergic neurons.

## Results

### RPE-like dopamine transient in the NAc in response to VTA dopamine neuron stimulation

Previous work established an optogenetics-powered Pavlovian conditioning task (hereon called opto-Pavlovian) wherein animals learn to associate the delivery of a cue with optogenetic activation of their midbrain dopamine neurons (*Saunders et al., 2018*). This previous study determines that dopaminergic neuron responses to optical stimulation-predictive cues become established over multiple learning sessions. However, in light of recent evidence demonstrating that dopamine release in the mesolimbic system and dopamine neuron activity can be uncoupled, we sought to determine whether

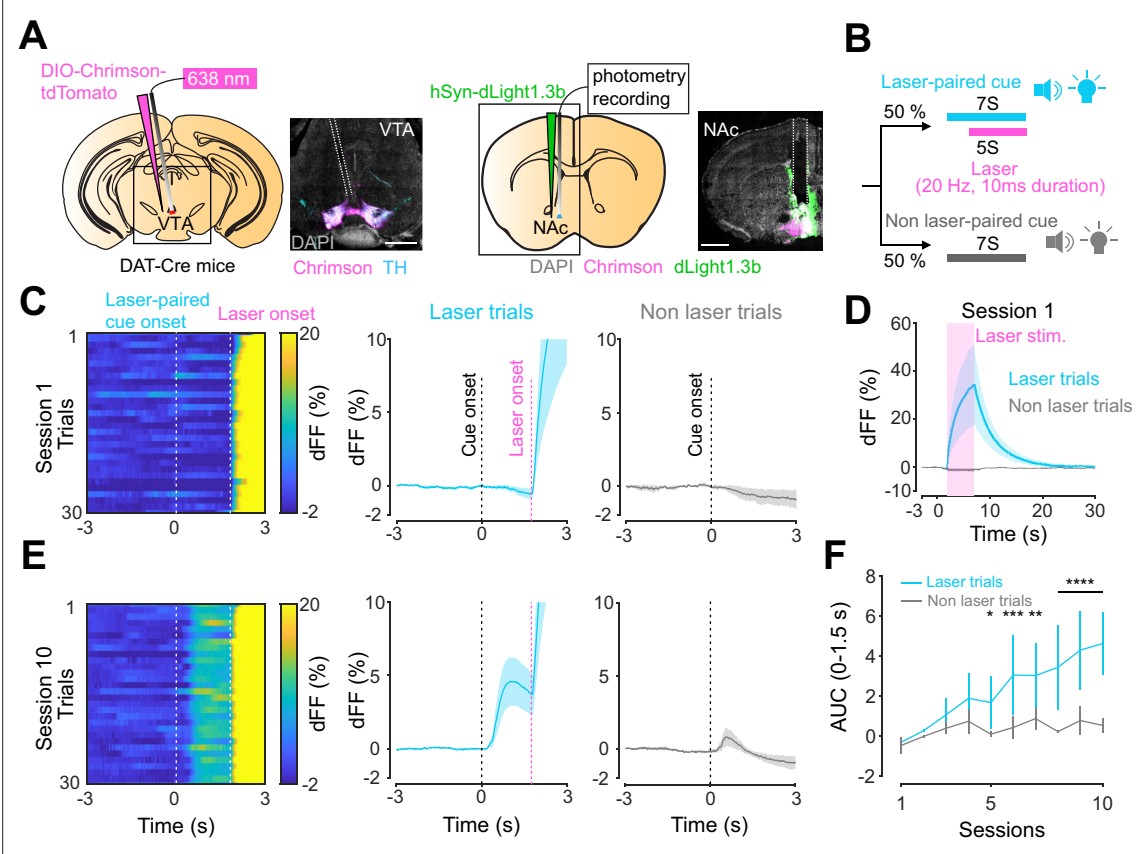

**Figure 1.** Dopamine transients in nucleus accumbens during an opto-Pavlovian task. (**A**) Preparation for opto-Pavlovian task combined with dLight recordings in the nucleus accumbens (NAc). Scale bar: 1 mm. White dashed lines indicate fiber tracts. (**B**) Schematic for opto-Pavlovian task. One cue was associated with the laser delivery while the other cue was not. (**C**) dLight recordings in the NAc of a representative mouse around the laser-paired cue presentation at session (left) and grouped data (middle). dLight recordings of non-laser-paired trials are also shown (right) at session 1. (**D**) dLight signals at session 1 during laser stimulation. The signals during non-laser trials are also shown. (**E**) The signals of a representative mice around laser-paired cue (left), grouped data (middle), and signals around non-laser-paired cue presentation (right) at session 10. (**F**) Area under the curve (AUC) of dLight signal in the NAc around the cue presentations (0–1.5 s) across sessions. Laser-paired cue triggered bigger transient than non-laser-paired cue. Two-way repeated-measures ANOVA. Session, $F_{9, 27} = 3.339$, p=0.0072. Cue, $F_{1, 3} = 3.997$, p=0.139. Interaction, $F_{9, 27} = 5.287$, p=0.0003. Tukey's multiple comparison, *p<0.05, **p<0.01,***p<0.001, and ***p<0.0001. n = 4 mice.

The online version of this article includes the following source data and figure supplement(s) for figure 1:

**Source data 1.** Source data for *Figure 1*.

**Source data 2.** Source Data for *Figure 1—figure supplement 1*.

**Figure supplement 1.** dLight recordings in the nucleus accumbens (NAc) during non-laser-paired cue delivery at sessions 1 (left) and 10 (right).

dopamine release would also follow the same patterns of dopamine somatic activity during this task (*Mohebi et al., 2019*; *Liu et al., 2022*). To selectively stimulate and monitor dopamine release from VTA dopaminergic neurons in the NAc, we injected a cre-dependent ChrimsonR AAV in the VTA as well as dLight1.3b (*Patriarchi et al., 2018*), a genetically encoded dopamine sensor AAV, in the NAc of DAT-cre mice. The recording optic fiber was placed directly above the NAc injection site (*Figure 1A*). Mice then underwent the 'opto-Pavlovian task' (*Saunders et al., 2018*), where one cue (tone + light, 7 s) was paired with the optogenetic stimulation of dopamine neurons in the VTA (*Figure 1D*), while the other cue was not (*Figure 1B*, *Figure 1—figure supplement 1*). We observed a gradual increase in dopamine transients in response to the delivery of the laser-associated cue (*Figure 1C, E, and F*). In contrast, the change in response to the non-laser-paired cue was smaller (*Figure 1C, E, and F*), suggesting that mice discriminated between the two cues. After 10 sessions of the opto-Pavlovian task, mice were exposed to omission sessions (*Figure 2A*), in which one-third of the laser-paired cues failed to trigger laser stimulation and the other two-thirds were followed by laser stimulation

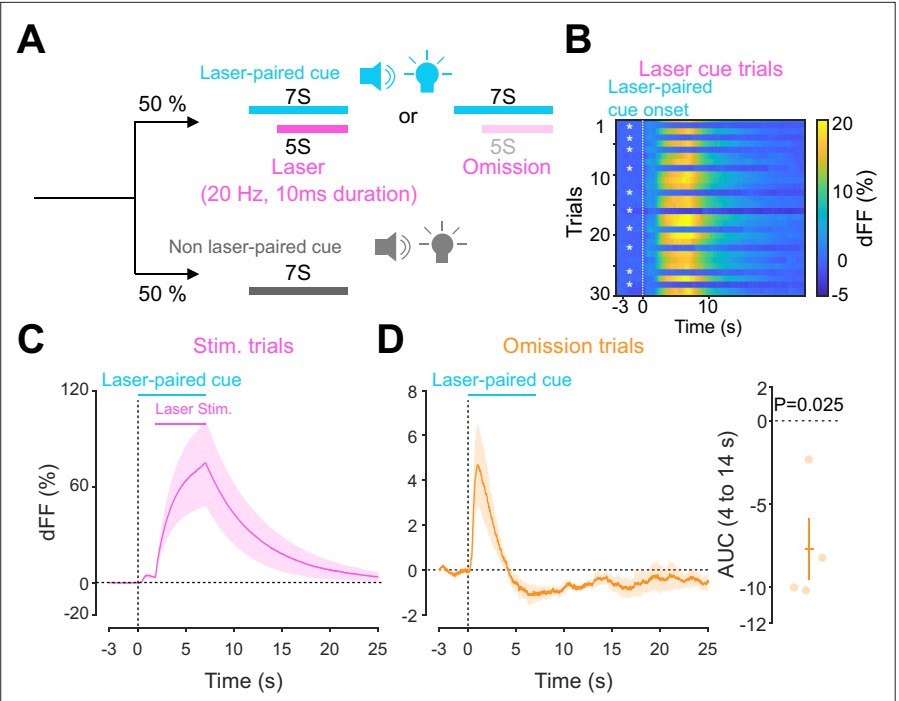

**Figure 2.** Accumbal dopamine transients during opto-Pavlovian omission trials. (**A**) Schematic for the omission sessions. Two-thirds of laser-associated cue were followed by the laser stimulation while the other one-third of the laser-associated cue failed to trigger the laser stimulation. (**B**) dLight recordings of a representative mouse during omission sessions. dLight signal around the laser-paired cue presentation is shown here. White asterisks indicate omission trials, while in the other trials, the laser stimulation was delivered. (**D**) dLight recordings in the nucleus accumbens (NAc) during stimulation trials and during omission trials (**C**). A dip of dLight signals was observed. One-sample *t*-test; *t* = 4.176, df = 3. p=0.0250. n = 4 mice.

The online version of this article includes the following source data for figure 2:

**Source data 1.** Source Data for *Figure 2*.

of VTA dopamine neurons (*Figure 2A–C*). The omission of the laser stimulation triggered a dip in dLight signal (*Figure 2D*). We also observed a small dip in dLight signal during non-laser-paired cue delivery (*Figure 1—figure supplement 1*). Overall, the dopamine transient observed during the opto-Pavlovian task was consistent with classical Pavlovian conditioning (*Saunders et al., 2018*; *Cohen et al., 2012*), indicating that mice engage in similar learning processes whether the reward consists of an edible entity or of optogenetic stimulation of VTA dopamine neurons.

### Dopamine transients in the LH follow the same rules as in the NAc

Given the involvement of the LH in reward-seeking behaviors (*Gibson et al., 2018*; *Nieh et al., 2015*), we next asked whether a similar neuromodulatory coding of predictive cues could take place in the hypothalamus, outside of the mesolimbic dopamine system. To answer this question, we followed the same procedure as for the NAc, except injecting dLight1.3 and positioning the optic fiber for photometry recordings in the LH (*Figure 3A*). We observed Chrimson-positive fibers in the LH originating from the VTA (*Figure 3A*) and found that the stimulation of VTA dopamine neurons reliably evoked dopamine transients in the LH (*Figure 3B*). The injected mice expressing dLight1.3b in the LH then underwent the opto-Pavlovian task (*Figure 3C–G*). In session 1 of the task, we observed dopamine transients neither around laser-paired cue nor around non-laser-paired cue presentation (*Figure 3C and D*). However, in the LH as in the NAc, there was a gradual increase in dopamine transients around the laser-paired cue delivery (*Figure 3E–G*), consistent with RPE-like dopamine transients. Omission sessions after 10 sessions of the task (*Figure 3H*) showed a dip in dopamine signal during omission trials (*Figure 3H*). These results are indicative of the presence of a certain amount of tonic dopamine in the LH under unstimulated conditions and that negative RPEs can induce a decrease in the

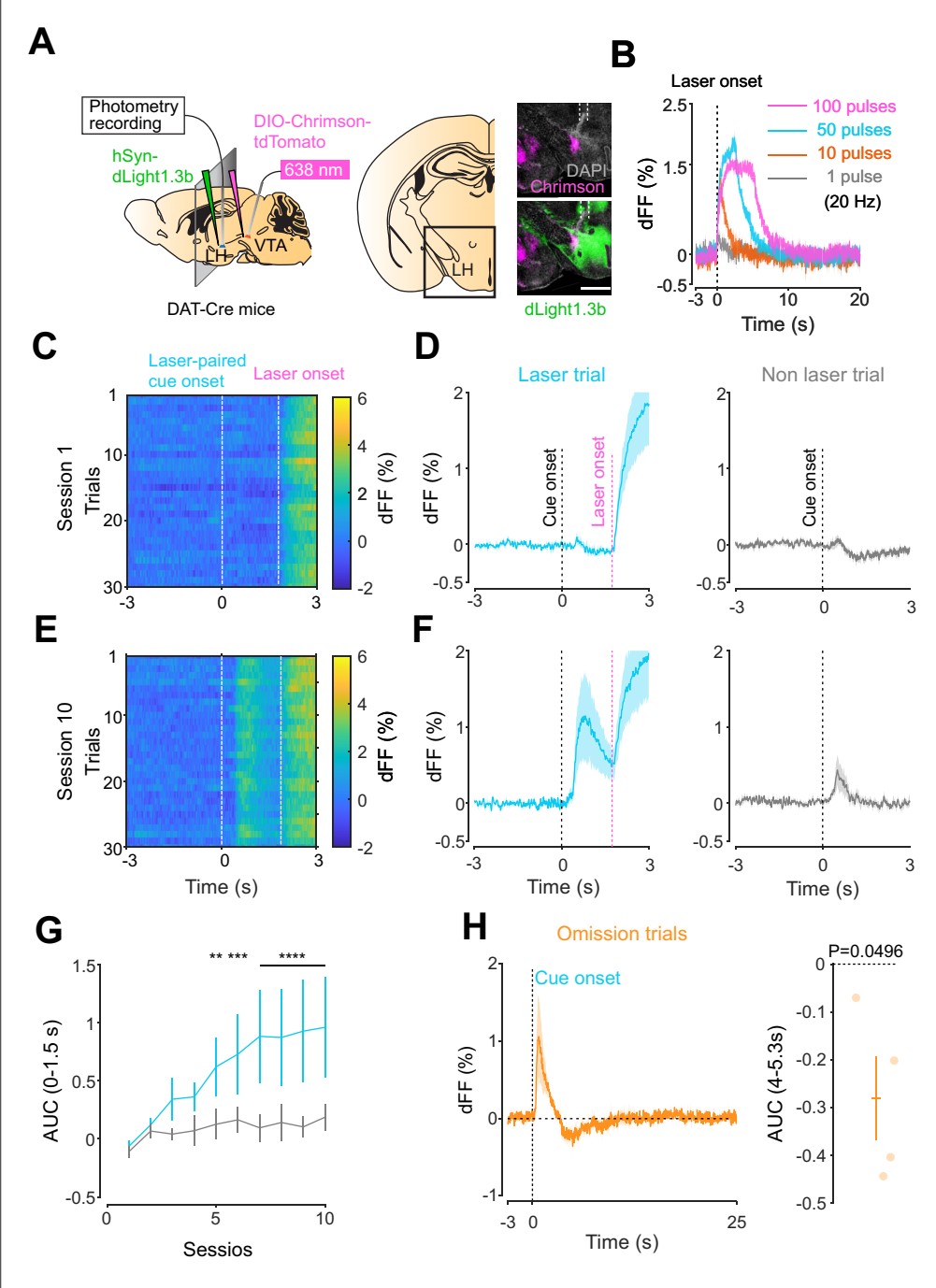

**Figure 3.** Mesohypothalamic dopamine dynamics associated with the opto-Pavlovian task. (**A**) Schematic for the dLight recording in the lateral hypothalamus (LH) while stimulating dopamine neurons in the ventral tegmental area (VTA) (left). Coronal image of the LH of a mouse infected with AAV-hSyn-DIO-Chrimson-tdTomato in the VTA and AAV-hSyn-dLight1.3b in the LH (right). White dashed lines indicate fiber tracts. Scale bar: 1 mm. (**B**) dLight signal in the LH during dopaminergic stimulation in the VTA at several number of pulses (20 Hz, 10 ms duration for each pulse). (**C**) dLight recordings during the laser-paired cue presentation of a representative mouse at session 1. (**D**) dLight recordings around the laser-paired cue presentation (left) and non-laser-paired cue presentation (right) at session 1. (**E**). dLight recordings during the laser-paired cue presentation of a representative mouse at session 10. (**F**). dLight recordings around the laser-paired cue presentation (left) and non-laser-paired cue presentation (right) at session 10. (**G**) Area under the curve (AUC) of dLight signal in the LH around the cue presentations (0–1.5 s) across sessions. Laser-paired cue triggered bigger transient than non-laser-paired cue. Two-way repeated-

*Figure 3 continued on next page*

*Figure 3 continued*

measures ANOVA. Session, $F_{9,27} = 3.814$, p=0.0033. Cue, $F_{1,3} = 5.818$, p=0.0948. Interaction, $F_{9,27} = 3.923$, p=0.0027. Tukey's multiple comparison, *p<0.05, **p<0.01,***p<0.001, and ***p<0.0001. (**H**) dLight recordings in the LH during omission trials. A dip in dLight signals was observed. One-sample *t*-test; *t* = 3.193, df = 3. p=0.0496. n = 4 mice.

The online version of this article includes the following source data for figure 3:

**Source data 1.** Source Data for *Figure 3*.

concentration of LH dopamine. Interestingly, the dopamine transients in the LH observed in these experiments mirrored the RPE-encoding dopamine responses we observed in the NAc.

## Different kinetics of dopamine in the NAc and LH

After conducting dLight recordings in the NAc and LH during the opto-Pavlovian task, we observed distinct kinetics of dopamine in these two brain regions. First, we compared the dopamine transient during stimulation trials of omission sessions, where mice already learned the association between the cue and the laser stimulation (*Figure 4A*). In the NAc, the dLight signal continued to increase until the laser was turned off, while in the LH, the dLight signal plateaued shortly after the initiation of the laser stimulation (*Figure 4A*). To precisely assess the kinetics of the dLight signals, we calculated their temporal derivatives (*Figure 4B*). In the NAc, the derivative crossed zero shortly after the termination of the laser stimulation, while in the LH, the zero-crossing point was observed during the laser stimulation (*Figure 4B and C*), indicating a different timing of direction change in the dLight signal. We applied the same analysis to the omission trials (*Figure 4D–F*). Following the initiation of the laser-paired cue, two zero-crossing points of the derivative of the dLight signal were identified. The first one corresponded to the maximum of the dLight signal, and the second one corresponded to the minimum of the dLight signal. In the LH, both zero-crossing points were smaller than in the NAc, suggesting that LH dopamine exhibits faster kinetics.

## Orexin neuron dynamics during the opto-Pavlovian task

We next addressed the hypothesis positing that dopamine in the LH can modulate orexinergic neuronal activity. We injected DAT-cre mice with an orexin promoter-driven GCaMP6s (*Bracey et al., 2022*; *Viskaitis et al., 2022*; *Li et al., 2022*; *González et al., 2016b*), which has been reported to target orexin neurons with >96% specificity (*González et al., 2016b*), in the LH and used fiber photometry to monitor the calcium transients of LH orexinergic neurons while optically controlling dopamine release via ChrimsonR expressed in the VTA (*Figure 5A and B*). After the mice fully recovered from the surgery, they underwent the opto-Pavlovian task. In session 1, calcium transients in orexin neurons were not modulated by the presentation of laser-paired or non-laser-paired cues (*Figure 5C*), although laser stimulation triggered the increase in calcium signal (*Figure 5—figure supplement 1*). As we observed with dLight recordings in the NAc and LH, the orexin-specific GCaMP signal increased across sessions around the presentation of the laser-paired cue (*Figure 5D and E*), therefore following a similar time course to the evolution of dopamine release in the LH. After mice learned the association, we tested the omission of laser stimulation (*Figure 5F*). Unlike dopamine signals, we did not observe a dip in orexin activity during omission trials (*Figure 5F*). Orexin neuron activity is known to be associated with animal locomotion (*Karnani et al., 2020*; *Donegan et al., 2022*). To exclude the possibility that the increase in calcium signaling during laser-paired cue trials is an indirect effect of stimulation-induced locomotion (*Karnani et al., 2020*; *Donegan et al., 2022*), we performed photometry recordings and optogenetic stimulation of VTA dopaminergic terminals in the LH both in freely moving or in isoflurane-anesthetized conditions (*Figure 6A*). In both conditions, we observed an increased orexinergic neuron activity after the onset of laser stimulation (*Figure 6B and C*), suggesting that the observed upregulation in orexinergic neuronal activity is independent of animal locomotion. Finally, to identify which dopamine receptor is responsible for this increase in orexinergic calcium, we systemically (I.P.) injected a D1 (SCH 23390) or D2 (raclopride) receptor antagonist, and optically stimulated dopaminergic terminals in the LH (*Figure 6E*, *Figure 6—figure supplement 1*). Raclopride largely reduced the observed orexin neuronal activity increases while SCH 23390 did not, indicating that the signal is at least in part mediated by the D2 receptor (*Figure 6F*). Our experiments suggest that LH orexin neurons participate in the LH response to VTA dopamine, and

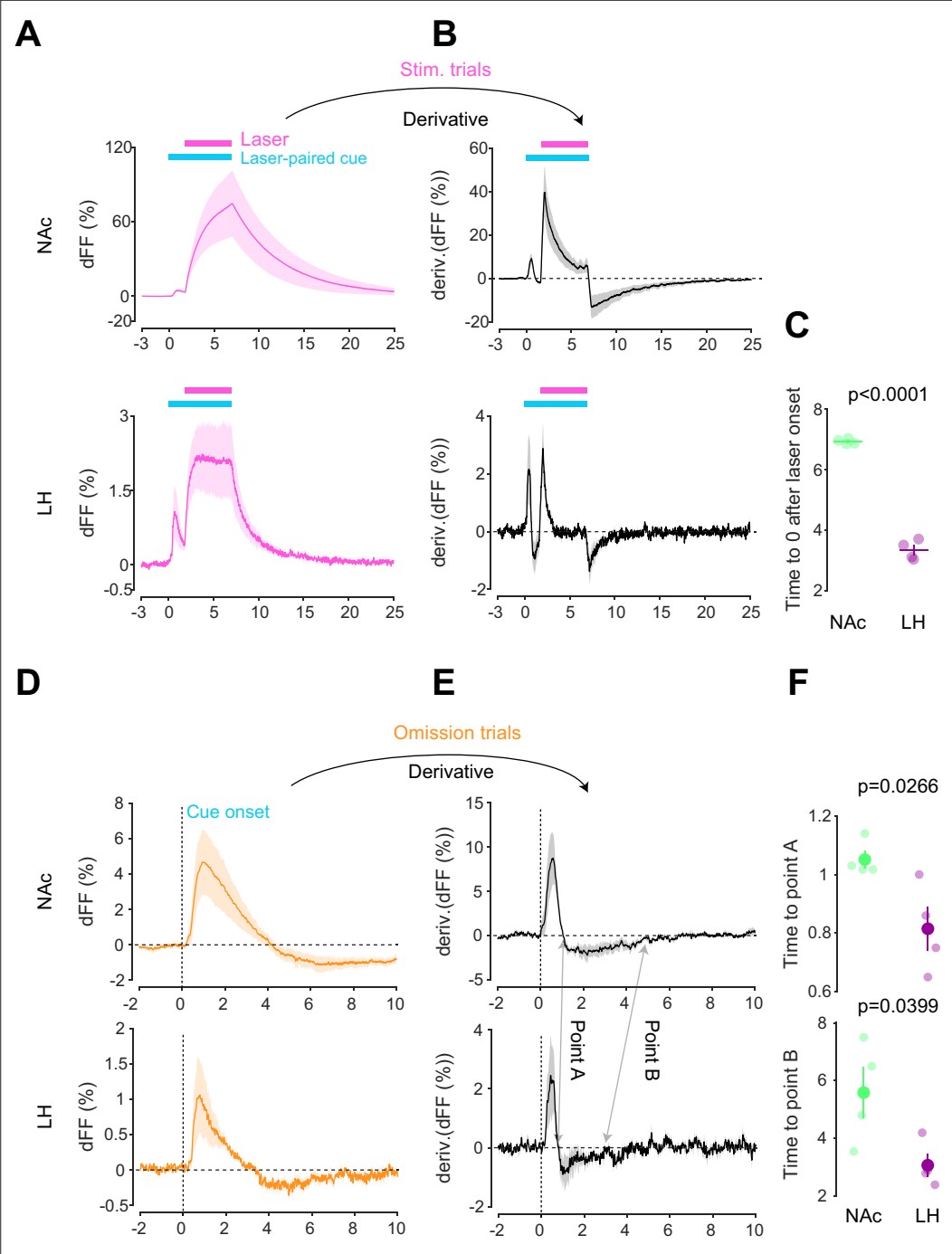

**Figure 4.** Kinetic differences in dopamine transients between mesoaccumbens and mesohypothalamic pathways. (**A**) dLight recordings in the nucleus accumbens (NAc) (top) and lateral hypothalamus (LH) (bottom) during optical stimulation of ventral tegmental area (VTA) dopamine neurons. (**B**) Derivative of panel (**A**). (**C**) quantification of zero-crossing point in panel (**B**) after the initiation of laser stimulation. Unpaired *t*-test; *t* = 21.69, df = 6. p<0.0001. (**D**) dLight recordings in the NAc (top) and LH (bottom) during omission trials. (**E**) Derivative of panel (**D**). (**F**) Quantification of first (top, point A) and second (bottom, point B) zero-crossing points after the initiation of the cue in panel (**E**). Top, unpaired *t*-test. *t* = 2.920, df = 6. p=0.0266. bottom, unpaired *t*-test. *t* = 2.614, df = 6. p=0.0399. Note that panels (**A**) and (**D**) are shown in *Figures 2 and 3* also. They are displayed for comparison purposes.

The online version of this article includes the following source data for figure 4:

**Source data 1.** Source Data for *Figure 4*.

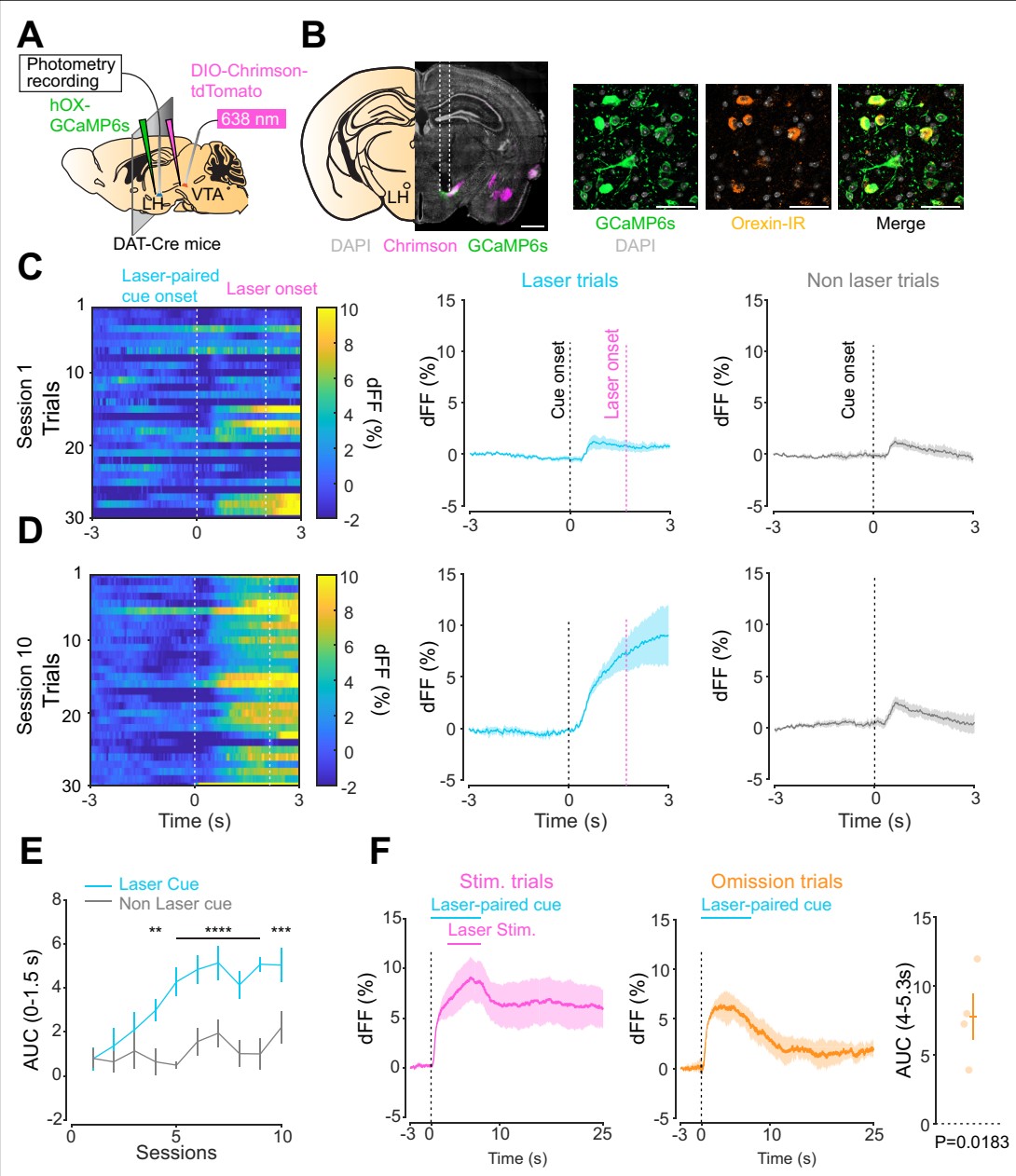

**Figure 5.** Orexin neuronal activity during an opto-Pavlovian task. (**A**) Schematic of the preparation for opto-Pavlovian task combined with orexin promoter GCaMP recordings in the lateral hypothalamus (LH). (**B**) Coronal image of a mouse brain slice infected with AAV-hSyn-DIO-ChrimsonR-tdTomato in the ventral tegmental area (VTA) and AAV1-hOX-GcaMP6S in the LH (left; scale bar; 1 mm). White dashed lines indicate fiber tracts. Zoom of infected LH with AAV1-hOX-GcaMP6s and co-localization orexin IR and GcaMP6s (right; scale bars; 50 µm). (**C**) Orexin promoter GcaMP recordings in the LH of a representative mouse around the laser-paired cue presentation at session 1 (left), grouped data (middle) and recordings during non-laser-paired trial (right). (**D**) Orexin promoter GcaMP recordings in the LH of a representative mouse around the laser-paired cue presentation at session 10 (left), grouped data (middle), and recordings during non-laser trial (right). (**E**) Area under the curve (AUC) of hOX-GcaMP signal in the LH around the cue presentations (0–1.5 s) across sessions. Laser-paired cue triggered bigger transient than non-laser-paired cue. Two-way repeated-measures ANOVA. Session, $F_{9, 27} = 4.438$, p=0.0012. Cue, $F_{1, 3} = 25.41$, p=0.0151. Interaction, $F_{9, 27} = 4.125$, p=0.0020. Tukey's multiple comparison, *p<0.05, **p<0.01, ***p<0.001, and ***p<0.0001. (**F**) Orexin promoter GCaMP recordings during stimulation trials (left) and omission trials (middle and right). AUC around the omission was higher than baseline. One-sample *t*-test; *t* = 4.693, df = 3. p=0.0183. n = 4 mice.

The online version of this article includes the following source data and figure supplement(s) for figure 5:

**Source data 1.** Source Data for *Figure 5*.

**Figure supplement 1.** Orexin-promoter GCaMP recording (left) and dLight recording during stimulation at session 1.

**Figure supplement 1—source data 1.** Source Data for *Figure 5—figure supplement 1*.

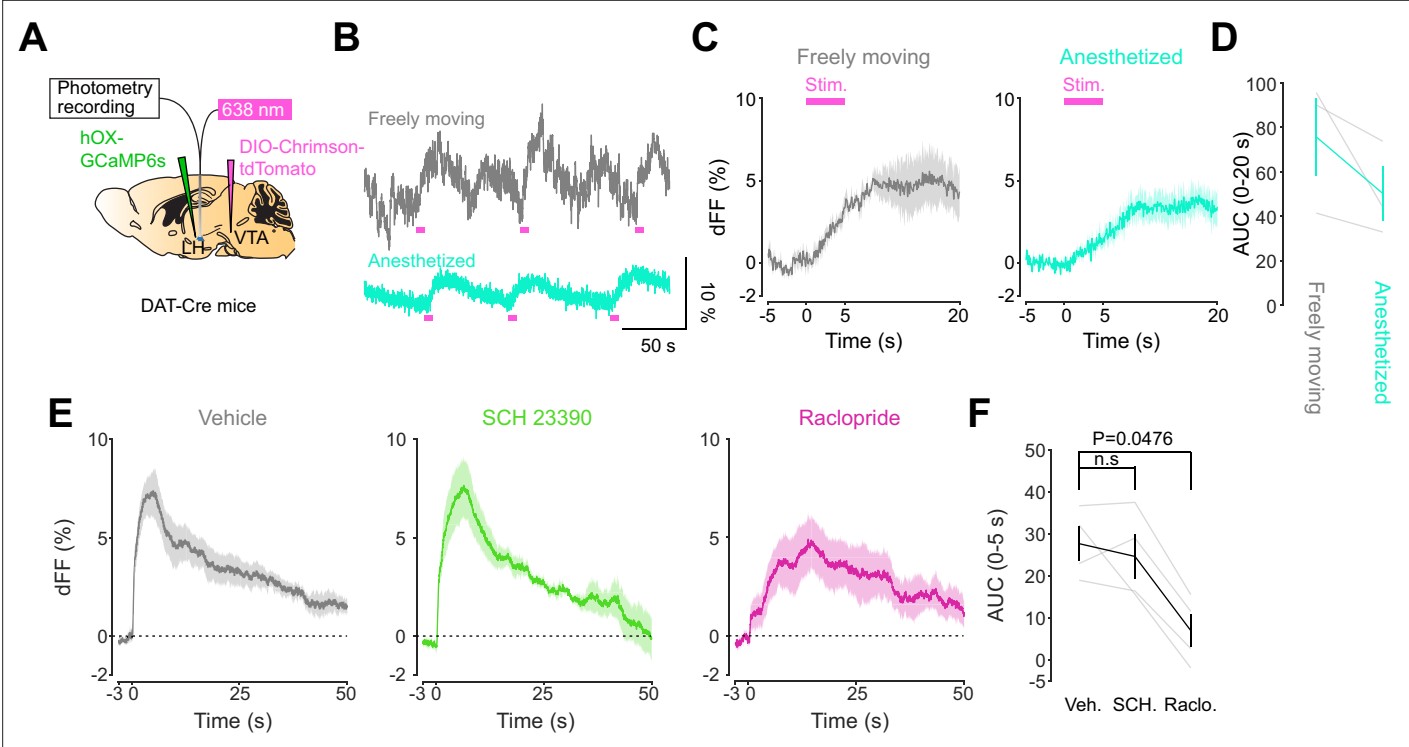

**Figure 6.** DA-dependent modulation of orexin neuronal activity is dependent on DRD2. (**A**) Schematic for the orexin promoter GCaMP recording in the lateral hypothalamus (LH) while stimulating dopamine terminals in the LH. (**B**) Orexin promoter GCaMP signals of a representative mouse. Recordings were performed while mice were freely moving (top) and anesthetized with isoflurane (bottom). Red bars indicate the stimulation (20 Hz, 100 pulses, 10 ms duration). (**C**) Orexin promoter GCaMP signals around the stimulation of dopamine terminals in the LH while animals were freely moving (left) and anesthetized (right). (D) Area under the curve (AUC) at 0–20 s was not significantly different between freely moving and anesthetized conditions. Paired *t*-test, t = 1.923, df = 2. p=0.1944. n = 3 mice. (**E**) In freely moving condition, recordings were performed after mice received the intraperitoneal injection of vehicle (left), SCH 23390 (1 mg/kg, middle), and raclopride (1 mg/kg, right). (**F**) AUC at 0–5 s. Black line indicates the mean for each condition and gray lines show individual mice. The administration of raclopride decreased the AUC significantly while SCH 23390 did not change the AUC. One-way ANOVA; F (3, 6) = 5.305, p=0.04. Tukey's multiple comparison test. vehicle vs. SCH 23390; p=0.8145. vehicle vs. raclopride; p=0.0476. n = 4 mice.

The online version of this article includes the following source data and figure supplement(s) for figure 6:

**Source data 1.** Source Data for *Figure 6*.

**Figure supplement 1.** Individual traces for *Figure 6E*.

that D2 receptors play an important role locally in the LH in regulating orexin neuron activity evoked by dopamine release.

## Discussion

The mesolimbic dopamine system has been proposed to encode RPEs (***Kim et al., 2020***; ***Cohen et al., 2012***; ***Schultz et al., 1997***), which signal a discrepancy between expected and experienced rewards. Recently, it has been demonstrated that the optical stimulation of midbrain dopamine neurons is sufficient to create Pavlovian conditioning (***Saunders et al., 2018***). While it is known that cells within the LH express several different dopamine receptor subtypes (***Yang et al., 2019***), and microinjection of D1 and D2 receptor agonists have been shown to decrease food intake in rodents (***Yonemochi et al., 2019***), before our study, dopamine transients in the LH during reward-associated tasks had not been reported. Here, we used an opto-Pavlovian task that echoed, with NAc dopamine measurements, already reported findings on the midbrain dopamine neurons' RPE-encoding role (***Saunders et al., 2018***). Then, we determined that VTA dopaminergic neurons release dopamine in the LH and found that dopamine transients in the LH in response to the same opto-Pavlovian task were qualitatively similar to those observed in the mesolimbic dopamine system.

Recent findings suggest that dopaminergic transients in the dorsal bed nucleus of the stria terminalis encode RPE (*Gyawali et al., 2023*), indicating qualitative similarities in dopamine activity within this brain region compared to what we observed in the LH and NAc. Conversely, dopamine responses in other brain regions, such as the medial prefrontal cortex (*Vander Weele et al., 2018*; *Verharen et al., 2020*) and amygdala (*Zhuo et al., 2023*; *Lutas et al., 2019*), predominantly react to aversive stimuli. Furthermore, we have found that dopamine in the LH also encodes RPE. However, the specific response of dopamine in the LH to aversive stimuli has not been fully explored, despite existing reports of significant orexinergic activity in response to such stimuli (*Yamashita et al., 2021*). This gap highlights the need for a detailed examination of how dopamine behaves in the LH when faced with aversive stimuli.

Indeed, during the opto-Pavlovian task, in which we stimulated VTA dopamine neurons and measured dopamine, we observed dopamine transients around a Pavlovian laser-paired cue presentation. We also observed a dip in dLight signal during omission trials, suggesting that a detectable concentration of dopamine is at extrasynaptic space in the LH at basal condition and that at the moment of omission the concentration of extrasynaptic dopamine decreases. These data indicate that dopamine transients in the LH, as in the NAc, could be encoding RPE.

While smaller than the response to the laser-paired cue, we observed modulation of the dLight signal in the NAc during the presentation of the non-laser-paired cue. In session 1, the cue presentation immediately triggered a dip, whereas in session 10, it evoked a slight increase in the signal, followed by a dip. Our hypothesis suggests that two components contribute to the dip in the signal. The first is the aversiveness of the cue; the relatively loud sound (90 dB) used for the cue could be mildly aversive to the experimental animals. Previous studies have shown that aversive stimuli induce a dip in dopamine levels in the NAc, although this effect varies across subregions (*Yang et al., 2018*; *Verharen et al., 2020*). The second component is related to RPE. While the non-laser-paired cue never elicited the laser stimulation, it shares similarities with the laser-paired cue in terms of a loud tone and the same color of the visual cue (albeit spatially different). We posit that it is possible that the reward-related neuronal circuit was slightly activated by the non-laser-paired cue. Indeed, a small increase in the signal was observed on day 10 but not on day 1. If our hypothesis holds true, as this signal is induced by two components, further analysis unfortunately becomes challenging.

While dopaminergic transients in the NAc and LH share qualitative similarities, the kinetics of dopamine differs between these two brain regions. Under optical stimulation, the dLight signal in the NAc exhibited a continuous increase, never reaching a plateau until the laser was turned off. In contrast, in the LH, the dLight signal reached a plateau shortly after the initiation of the laser stimulation. The distinction in dopamine kinetics was also evident during omission trials, where the dopamine kinetics in the LH were faster than those in the NAc. The molecular mechanisms underlying this difference in kinetics and its impact on behavior remain to be elucidated. Due to this kinetic difference, we employed distinct time windows to capture the dip in the dLight signal during omission trials.

Previous work indicates that orexin neurons project to VTA dopamine neurons (*Borgland et al., 2006*; *Thomas et al., 2022*; *Baimel et al., 2017*), facilitating dopamine release in the NAc and promoting reward-seeking behavior. However, while it has been demonstrated that systemic injection of dopamine receptor agonists activates orexin neurons (*Bubser et al., 2005*), their reciprocal connection with dopaminergic neurons had not yet been investigated in vivo (*Linehan et al., 2019*). Here, we studied the relationship between orexinergic and dopaminergic activity in the LH and found that LH dopamine transients and orexinergic neuronal activities are positively correlated. Seeing as dopamine-related orexinergic activity was reduced by systemic injections of raclopride, we postulate that dopamine in the LH activates orexin neurons via D2R. D2R couples to Gi proteins (*Ford, 2014*), so it is unlikely that dopamine directly activates orexin neurons. Our testable hypothesis is that dopamine modulates orexin neuron activation via a disinhibitory mechanism; for example, GABA interneurons could be inhibited by the activation of D2R, consequently disinhibiting orexin neurons (*Ferrari et al., 2018*; *Burt et al., 2011*). It has been established that D1 receptor expressing medium spiny neurons (D1-MSNs) in the NAc densely project to the LH, especially to GABAergic neurons (*O'Connor et al., 2015*; *Thoeni et al., 2020*), raising a possibility that dopamine in the LH modulates the presynaptic terminals of D1-MSNs. However, administration of D1R antagonist (SCH 23390) did not block the calcium transient in orexin neurons evoked by the dopaminergic terminal stimulation in the LH, implying that the contribution of D1-MSNs to orexin neuronal activity is minimal in our experimental

design. While systemic injections of raclopride effectively reduced dopaminergic terminal stimulation-evoked orexinergic activity, the long-lasting calcium signal remained unaltered (*Figure 6E*). This discrepancy could arise from an insufficient blockade of dopamine receptors. For D1R blockade, we administered 1 mg/kg of SCH-23390 5 min before recordings. This dose is adequate to induce behavioral phenotypes (*Womer et al., 1994*) and block D1R-based dopamine sensors (*Patriarchi et al., 2018*), although higher doses have been used in some studies (*Zhuo et al., 2023*). To block D2R, we injected 1 mg/kg of raclopride, a dose known to induce hypo-locomotion (*Simón et al., 2000*), indicating effective modification of the neuronal circuit. However, these data do not guarantee complete receptor blockade, and it is possible that optical stimulation resulted in high extrasynaptic dopamine concentration, leading to partial receptor binding. Alternatively, this component might be mediated by other neurotransmitters, such as glutamate (*Mingote et al., 2017*; *Zell et al., 2020*; *Dal Bo et al., 2004*) or GABA (*Melani and Tritsch, 2022*), which are known to be co-released from dopaminergic terminals.

Several ex vivo experiments suggest that dopamine, particularly at high concentrations (50 μM or higher), reduces the firing rate of orexin neurons, albeit with a potency significantly lower than that of norepinephrine (*Yamanaka et al., 2006*; *Li and van den Pol, 2005*) through both direct and indirect mechanisms (*Linehan et al., 2015*; *Linehan et al., 2019*). This apparent discrepancy with our results could be attributed to a different time course of dopamine transients. In slice experiments, the concentration of exogenous dopamine or dopamine agonists is determined by the experimenter and often maintained at high levels for minutes. In contrast, in our experimental setup, dopamine evoked by laser stimulation is degraded/reuptaken as soon as the laser is turned off. This variation in the time course of dopamine transients could contribute to the observed differences in responses to dopamine. Another plausible explanation for this discrepancy is the difference in dopamine concentration. Modulations of synaptic transmission to orexinergic neurons by dopamine are reported to be concentration-dependent (*Linehan et al., 2015*). Despite the brightness of the genetically encoded dopamine sensor following a sigmoidal curve in response to changes in dopamine concentration (*Patriarchi et al., 2018*), estimating dopamine concentration in vivo based on the sensor's brightness is not technically feasible. Therefore, it is challenging to determine the exact dopamine concentration achieved by laser stimulation, and it is possible that this concentration differs from the one that triggers the reduction in the firing rate of orexin neurons.

Although presentation of laser-paired cue and laser stimulation of VTA dopamine neurons evoked dopamine transient in the LH and an increase in calcium signals of orexin neurons, we did not observe a dip in calcium signal of orexin neurons during omission trials. This lack of a dip could be due to (1) slow sensor kinetics (*Zhang et al., 2023*) – since the pre-omission cue triggers LH dopamine release, and increases the calcium transient in orexin neurons, if the kinetics of GCaMP6s expressed in orexin neurons were too slow, we would not be able to observe an omission-related orexin activity dip – and (2) dopamine signaling properties. Dopamine receptors couple to G proteins (*Baik, 2013*), which act relatively slowly, potentially preventing us from seeing an omission-related signaling dip. Both theories are compatible with our observation that orexinergic activity increases over time during the presentation of our laser-paired cue, as our observed increases are not sporadic but developed over time. Recent studies indicate that orexin neurons respond to cues associated with reward delivery. However, unlike dopaminergic responses, which linearly correlate with the probability of reward delivery, the orexin response plateaus at around 50% probability of reward delivery (*Bracey et al., 2022*). This observation indicates that orexin neurons encode multiplexed cognitive information rather than merely signaling RPE. Our data indicate a direct conveyance of dopaminergic information, specifically RPE, to orexinergic neurons. However, the mechanism by which orexinergic neurons process and convey this information to downstream pathways remains an open question.

The silencing of orexinergic neurons induces conditioned place preference (*Garau et al., 2020*), suggesting that the silencing of orexin neurons is positively reinforcing. Considering that the stimulation of VTA dopamine neurons (*Harada et al., 2021*; *Pascoli et al., 2018*) and dopaminergic terminals in the LH (*Hoang et al., 2023*) is generally considered to be positively reinforcing, the activation of orexin neurons by dopaminergic activity might be competing with dopamine's own positive reinforcing effect. At the moment of omission, we observed a dopamine dip both in the NAc and LH, while orexin neurons were still activated. These data suggest that there is a dissociation between dopamine concentration and orexin neuronal activity at the moment of omission. This raises the

intriguing possibility that this dissociation – the activation of orexin neurons during a quiet state of dopamine neurons – could be highly aversive to the mice, therefore could be playing a role in negative reinforcement (*Iino et al., 2020*; *Cohen et al., 2012*; *González et al., 2016a*).

It has been demonstrated that the orexin system plays a critical role in motivated learning (*Sakurai, 2014*). Blocking orexin receptors impairs Pavlovian conditioning (*Keefer et al., 2016*), operant behavior (*Sharf et al., 2010*), and synaptic plasticity induced by cocaine administration (*Borgland et al., 2006*). Additionally, dopamine in the LH is essential for model-based learning, and the stimulation of dopaminergic terminals in the LH is sufficient to trigger reinforcement learning (*Hoang et al., 2023*). These collective findings strongly suggest that the activation of orexin neurons, evoked by dopamine transients, is crucial for reinforcement learning. Our data indicate that dopamine in both the NAc and LH encodes RPE. One open question is the existence of such a redundant mechanism. We hypothesize that dopamine in the LH boosts dopamine release via a positive feedback loop between the orexin and dopamine systems. It has already been established that some orexin neurons project to dopaminergic neurons in the VTA, positively modulating firing (*Thomas et al., 2022*). On the other hand, our data indicate that dopamine in the LH stimulates orexinergic neurons. These collective findings suggest that when either the orexin or dopamine system is activated, the other system is also activated consequently, followed by further activation of those systems. Although the current findings align with this idea, the hypothesis should be carefully challenged and scrutinized.

In summary, by implementing an opto-Pavlovian task combined with fiber photometry recordings, we found evidence that the meso-hypothalamic dopamine system exhibits features qualitatively similar to those observed in the mesolimbic dopamine system – where dopamine is thought to encode RPEs. Furthermore, our findings show that dopamine in the LH positively modulates the neuronal activities of orexin neurons via D2 receptors. These findings give us new insights into the reciprocal connections between the orexin and dopamine systems and shed light on the previously overlooked direction of dopamine to orexin signaling, which might be key for understanding negative reinforcement and its dysregulation.

## Materials and methods

### Animals

All animal procedures were performed in accordance with the Animal Welfare Ordinance (TSchV 455.1) of the Swiss Federal Food Safety and Veterinary Office and were approved by the Zurich Cantonal Veterinary Office. Adult DAT-IRES-cre mice (B6.SJL-Slc6a3tm1.1(cre)Bkmn/J; Jackson Labs), referred to as Dat-cre in the article, of both sexes were used in this study. Mice were kept in a temperature- and humidity-controlled environment with ad libitum access to chow and water on 12 hr/12 hr light/dark cycle.

### Animal surgeries and viral injections

Surgeries were conducted on adult anesthetized mice (males and females, age >6 wk). AAV5-hSyn-FLEX-ChrimsonR-tdTomato (UNC Vector Core, 7.8x10E12 vg/ml) was injected in the VTA (–3.3 mm AP, 0.9 mm ML, –4.28 mm DV, with 10° angle, volume 600 nl). Above the injection site, a single optic fiber cannula (diameter 200 μm) was chronically implanted (–3.3 mm AP, 0.9 mm ML, –4.18 mm DV). In the NAc (1.5 mm AP, 0.7 mm ML, –4.5 mm DV), AAV9-hSyn1-dLight1.3b-WPRE-bGHp (Viral Vector Facility,7.9 × 10E12 vg/ml) was injected and an optic fiber (diameter 400 μm) was implanted (1.5 mm AP, 0.7 mm ML, –4.4 mm DV) for photometry recordings. In some mice, dLight virus or AAV1.pORX. GCaMP6s.hGH (*Bracey et al., 2022*) was injected in the LH (–1.4 mm AP, 1.1 mm ML, –5.0 mm DV), followed by an optic fiber implantation (–1.4 mm AP, 1.1 mm ML, –4.8 mm DV).

### Opto-Pavlovian task

Dat-cre mice infected with AAV5-hSyn-FLEX-ChrimsonR-tdTomato in the VTA were placed in an operant chamber inside a sound-attenuating box with low illumination (30 Lux). Chamber functions synchronized with laser light deliveries were controlled by custom-written MATLAB scripts via a National Instrument board (NI USB 6001). The optic fiber implanted above the VTA was connected to a red laser (638 nm, Doric Lenses; CLDM_638/120) via an FC/PC fiber cable (M72L02; Thorlabs) and a simple rotary joint (RJ1; Thorlabs). Power at the exit of the patch cord was set to 15 ± 1 mW. Two

visual cues were in the operant chamber and a speaker was placed inside the sound-attenuating box. The laser-predictive cue was composed of the illumination of one visual stimulus (7 s continuous) and a tone (5 kHz, 7 s continuous, 90 dB), while the non-laser-paired cue was composed of a second visual stimulus (7 s continuous) and a different tone (12 kHz, 7 s continuous, 90 dB). Each cue was presented for 7 s. Two seconds after the onset of the laser-predictive cue, the red laser was applied for 5 s (20 Hz, 10 ms pulse duration). The presentation of the non-laser cue was followed by no stimuli. In random interval 60 s (45–75 s), one cue was presented in a pseudorandom sequence (avoiding the presentation of the same trials more than three times in a row). Mice were exposed to 30 laser cues and 30 non-laser-paired cues in each session. Mice were trained 5 d per week. After 10 sessions of opto-Pavlovian training, mice underwent two sessions of omission. In the omission sessions, two-thirds of laser-paired cue presentation were followed by the delivery of the laser stimulation (laser trial), and one-third of laser-paired cue presentation did not lead to laser stimulation (omission trial). The laser-paired cue was kept the same for laser and non-laser trials. Each omission session was composed of 20 laser trials, 10 omission trials, and 30 non-laser trials.

## Photometry recordings

Fiber photometry recordings were performed in all the sessions. Dat-cre mice injected with AAV9-hSyn1-dLight1.3b-WPRE-bGHp in the NAc or LH, or AAV1.pORX.GCaMP6s.hGH in the LH were used. All the mice were infected with AAV5-hSyn-FLEX-ChrimsonR-tdTomato in the VTA. iFMC6_IE(400-410)_E1(460-490)_F1(500-540)_E2(555-570)_F2(580-680)_S photometry system (Doric Lenses) was controlled by the Doric Neuroscience Studio software in all the photometry experiments except for the anesthesia experiment of *Figure 6*. In the experiment in *Figure 6*, a two-color+optogenetic stimulation rig (Tucker-Davis Technologies, TDT) was used. Mice were exposed to 5% isoflurane for anesthesia induction and were kept anesthetized at 2% isoflurane through the rest of the experiment. The recordings started 10 min after the induction of anesthesia. A low-autofluorescence patch cord (400 μm, 0.57 N.A., Doric Lenses) was connected to the optic fiber implanted above the NAc or LH. The NAc or LH was illuminated with blue (465 nm, Doric) and violet (405 nm, Doric) filtered excitation LED lights, which were sinusoidally modulated at 208 Hz and 572 Hz (405 nm and 465 nm, respectively) via lock-in amplification, then demodulated online and low-passed filtered at 12 Hz in the Doric System. In the TDT system, signals were sinusoidally modulated using the TDT Synapse software and an RX8 Multi I/O Processor at 210 Hz and 330 Hz (405 nm and 465 nm, respectively) via a lock-in amplification detector, then demodulated online and low-passed filtered at 6 Hz. Analysis was performed offline in MATLAB. To calculate ΔF/F0, a linear fit was applied to the 405 nm control signal to align it to the 470 nm signal. This fitted 405 nm signal was used as F0 in standard ΔF/F0 normalization $\{F(t) - F0(t)\}/F0(t)$. For the antagonist experiments in *Figure 5*, SCH-23390 (1 mg/kg in saline) or raclopride (1 mg/kg in saline) was injected (I.P.) 5 min before recordings.

## Immunohistochemistry

Perfused brains were fixed with 4% paraformaldehyde (Sigma-Aldrich) overnight (room temperature) and stored in PBS at 4°C for a maximum of 1 mo. Brains were sliced with a Vibratome (Leica VT1200S; feed = 60 μm, freq = 0.5, ampl = 1.5), and brain slices near the fiber tracts were subsequently selected for staining. These slices were permeabilized with 0.3% Triton X-100 for 10 min (room temperature). Next, they were incubated with blocking buffer for 1 hr (5% bovine serum albumin; 0.3% Triton X-100) before staining with the respective primary antibodies (NAc and LH with αGFP chicken 1:1000, Aves Labs ref GFP-1010; αmCherry rabbit, 1:1000, abcam ab167453; and αOrexin goat, 1:500, Santa Cruz Biotech, C-19; VTA with αmCherry rabbit, 1:1000, abcam, ab167453; and αTH chicken, 1:500, TYH0020) overnight. After three washes with 0.15% Triton, samples were incubated with the respective secondary antibodies and DAPI (for GFP donkey-α chicken, 1:1000, Alexa Fluor 488, 703-545-155; for mCherry donkey-α rabbit 1:67, Cy3, Jackson, 711-165-152; for orexin donkey-α goat, 1:500, Cy5; for TH donkey-α chicken, 1:67, Alexa Fluor647, 703-605-155; for DAPI 1:2000, Thermo Fisher, 62248) for 1 hr. Finally, samples were washed three times with PBS and mounted on microscope slides with a mounting medium (VectaShield HardSet with DAPI, H-1500-10). Image acquisition was performed with a ZEISS LSM 800 with Airyscan confocal microscope equipped with a Colibri 7 light source (Zeiss Apochromat).

## Statistical analysis

Statistical analysis was performed in GraphPad Prism9. For all tests, the threshold of statistical significance was placed at 0.05. For experiments involving one subject, one-sample $t$-test was used. For experiments involving two independent subjects or the same subjects at two different time points, two-tailed Student's unpaired or paired $t$-test was used, respectively. For experiments involving more than two groups, one-way or two-way ANOVA was performed and followed by Tukey's multiple comparison test. All data are shown as mean ± SEM.

## Acknowledgements

We acknowledge funding from the Swiss National Science Foundation (grant agreement no. 310030_196455) (TP), the European Union's Horizon 2020 research and innovation program (grant agreement no. 891959 to TP), and the University of Zürich. We would like to thank Jean-Charles Paterna and the Viral Vector Facility of the Neuroscience Center Zürich (ZNZ) for the kind help with virus production.

## Additional information

### Funding

| Funder | Grant reference number | Author |
| --- | --- | --- |
| Schweizerischer Nationalfonds zur Förderung der Wissenschaftlichen Forschung | 310030_196455 | Tommaso Patriarchi |
| European Research Council | 891959 | Tommaso Patriarchi |

The funders had no role in study design, data collection and interpretation, or the decision to submit the work for publication.

### Author contributions

Masaya Harada, Conceptualization, Data curation, Formal analysis, Investigation, Visualization, Methodology, Writing - original draft, Writing - review and editing; Laia Serratosa Capdevila, Investigation, Visualization, Methodology, Writing - original draft, Writing - review and editing; Maria Wilhelm, Methodology, Writing - review and editing; Denis Burdakov, Resources, Writing - original draft, Writing - review and editing; Tommaso Patriarchi, Conceptualization, Resources, Data curation, Supervision, Funding acquisition, Validation, Visualization, Writing - original draft, Project administration, Writing - review and editing

### Author ORCIDs

Masaya Harada http://orcid.org/0000-0003-2958-2772
Tommaso Patriarchi http://orcid.org/0000-0001-9351-3734

### Ethics

All animal procedures were performed in accordance to the Animal Welfare Ordinance (TSchV 455.1) of the Swiss Federal Food Safety and Veterinary Office and were approved by the Zurich Cantonal Veterinary Office. Adult DAT-IRES-cre (B6.SJL-Slc6a3tm1.1(cre)Bkmn/J; Jackson Labs) mice of both sexes were used in this study.

Reviewer #1 (Public Review): https://doi.org/10.7554/eLife.90158.3.sa1
Reviewer #3 (Public Review): https://doi.org/10.7554/eLife.90158.3.sa2
Author Response https://doi.org/10.7554/eLife.90158.3.sa3

# Additional files

## Supplementary files
• MDAR checklist

## Data availability
All data generated or analysed during this study are included in the manuscript and supporting files; source data files have been provided for Figures 1, 2, 3, 4, 5, 6.

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
