## [Editor Report · eLife assessment]

This study presents **valuable** findings that expand our view of dopamine release in different brain regions and show that dopamine release in the lateral hypothalamus is related to the activity of orexin neurons. The evidence supporting the claims of the authors is **solid**, although inclusion of tests that directly assess causality of the noble pathways would have been even more conclusive. The work will be of interest of neuroscientists who study the neural basis of motivation.

---

## [Referee Report · Reviewer #1 (Public Review)]

Summary:

Mice can learn to associate sensory cues (sound and light) with a reward or activation of dopamine neurons in the ventral tegmental area (VTA), and then anticipate the reward from the sensory cue only. Using this paradigm, Harada et al. showed that after learning, the cue is able to induce dopamine release in the projection targets of the VTA, namely the nucleus accumbens and lateral hypothalamus (LH). Within the LH, dopamine release from VTA neurons (either by presentation of the cue or direct optical stimulation of VTA neurons) activates orexin neurons, measured as an increase in intracellular calcium levels.

Strengths:

This study utilized genetically encoded optical tools to selectively stimulate dopamine neurons and to monitor dopamine release in target brain areas and calcium response of orexin neurons. This allowed a direct assessment of the relationship between the behavioral response of the animals, release of a key neurotransmitter in select brain areas and its effect on target cells with the precision previously not possible. The results shed light onto the mechanism underlying reward-related learning and expectation.

Weaknesses:

Supplementary Fig.2: While the differences in time course are analyzed and extensively discussed, there is also a large discrepancy in the magnitude of change in DA levels in the two areas that is not mentioned. Specifically, DA increases is about 90-fold of baseline in NAc while it is about 2-fold in the LH. This could be because the DA level is either higher during baseline or lower during peak in the LH. Is there a known difference in the DA fiber density or extracellular DA levels in the two areas?

The DA antagonist i.p. study (Fig.5E and suppl fig 4) appears to be repeated measurements in same animals. If so, is it possible that repeated opto-sessions result in desensitization of the response, and therefore the smaller response is not due to the antagonist? Ideally, the order of experiments (i.e. vehicle, SCH23390 and raclopride) would be randomized, and a control group should be shown where DA terminal-stimulation induces consistent response in orexin neurons when applied three times without any antagonists. The result should be assessed using one-way repeated measures ANOVA.

Importantly, only 5 minutes were allowed for i.p. injected drugs to be absorbed and distributed to the brain before DA release was evoked and ORX neuron activity were monitored. Unfortunately, this is too short (In Ref 13, ip injection of SCH 23390 was 30 minutes prior to optogenetics/photometry experiments. In Ref 70, no effect on behavior was detected at 10 min post-i.p. injection of SCH 23390; In Ref 71, the effect of raclopride on behavior was measured 30 min post-ip injection).

Overall, it seems premature to make a conclusion about a role for D2 receptors or lack of involvement of D1 receptors in the observed phenomenon.

Reciprocal activation of VTA DA neurons and LH orexin neurons is an interesting idea. However, if this is the case, the activity of these two types of cells should show similar pattern and time course. This manuscript shows that extracellular DA levels decays quickly following the cessation of optical stimulation (Fig. 3B) whereas orexin neuron activity is long-lasting (Fig. 5). Thus, the hypothesis does not seem to be fully supported by experimental data.

---

## [Referee Report · Reviewer #3 (Public Review)]

Summary:

Harada and colleagues describe an interesting set of experiments characterizing the relationship between dopamine cell activity in ventral tegmental area (VTA) and orexin neuron activity in lateral hypothalamus (LH). All experiments are conducted in the context of an opto-Pavlovian learning task, in which a cue predicts optogenetic stimulation of VTA dopamine neurons. With training, cues that predict DA stimulation come to elicit dopamine release in LH (a similar effect is seen in accumbens). After training, omission trials (cue followed by no laser) result in a dip (inhibition) of dopamine release in LH, characteristic of reward prediction error observed in striatum. Across cue training, the activity pattern of orexin neurons in LH mirrors that of LH DA levels. However, unlike the DA signal, orexin neurons do not exhibit a decrease in activity in omission trials. Systemic blockade of D2 but not D1 receptors blocked DA release in LH following VTA DA cell stimulation.

Strengths:

Although much work has been dedicated to examining projections from orexin cells to VTA, less has been done to characterize reciprocal projections and their function. In this way, this paper is a very important addition to the literature. The experiments are technically sound (with some limitations, below) and utilize sophisticated approaches, the manuscript is nicely written, and the conclusions are mostly reasonable based on the data collected.

Weaknesses:

I believe the impact of the paper could be enhanced by considering and/or addressing the following:

Major

• I encourage the authors to discuss in the Introduction previous work on DA regulation of orexin neurons. In particular, the authors cite, but do not describe in any detail, the very relevant Linehan paper (2019; Am J Physiol Regul) which shows that DA differentially alters excitatory/inhibitory input onto orexin neurons and that these actions are reversed by D1 vs D2 receptor antagonists. Another paper (Bubser, 2005, EJN) showed that dopamine agonists increase activity of orexin neurons and that these effects are blocked by D1/D2 antagonists. The current findings should be discussed in the context of these (and any other relevant) papers in the Discussion, too.

The revised manuscript addresses these concerns.

• In the Discussion, the authors provide 2 (plausible) explanations for why they did not observe a dip in calcium signal of orexin neurons during omission trials. Is it not possible that these cells do not encode for this type of RPE?

The revised manuscript addresses these concerns.

• Related to the above - I am curious about the authors' thoughts on why there is such redundancy in the system. i.e. why is dopamine doing the same thing in NAC and LH in the context of cue-reward learning?

The revised manuscript addresses these concerns.

• The data, as they stand, are largely correlative and do not indicate that DA recruitment of orexin neurons is necessary for learning to occur. It would be compelling if blocking the orexin cell recruitment affected some behavioral outcome of learning. Similarly - does raclopride treatment across training prevent learning?

I maintain that experiments testing the causality of these effects on learning/behavior would enhance the impact of the paper. However, I recognize that this would require substantial additional experimentation and the data here are interesting regardless.

• Only single doses of SCH23390 and raclopride were used. How were these selected? It would be nice to use more of a dose range to show that (1) and effect of D1R blockade was not missed, and (2) that the reduction in orexin signal with raclopride was dose-dependent.

Additional information on dose selection has been included - thank you. Again, these data might be more impactful if the effects of antagonists were found to be dose-dependent.

• Fig 1C, could the effect the authors observed due to movement? Relatedly, what was the behavior like when the cue was on? Did mice orient/approach the cue? Also, when does the learning about the cue occur? Does it take all 10 days of learning or does this learning/cue-induced increase in dopamine signaling occur in less than 10 days?

These have been addressed in the revised manuscript

• Also related to above, could the observed dopamine signal be a result of just the laser turning on? It would seem important to include mice with a control sensor.

The authors note that a control channel was recorded. I agree this is useful, but my concern is that the illumination of laser itself might startle the animal (promote movement), resulting in dopamine release. Showing this does not occur with the same laser in chr2-lacking vta neurons would help resolve this issue.

• Fig 1E, the effect seems to be driven by one mouse which looks like it could be a statistical outlier. Inclusion of additional animals would make these data more compelling.

I would still argue that these data could be strengthened by the addition of more mice. I note that the graph depicting individual data points has been removed from the revised manuscript - i would recommend re-including this figure.

• For Fig 1C, 3D, 3F, and 4D, could the authors please show the traces for the entire length of laser onset? It would be helpful to see both the rise and the fall of dopamine signals.

• Fig 2C, could the authors comment on how they compared the AUC to baseline? Was this comparison against zero? Because of natural hills and troughs during signals prior to cue (which may not equate to a zero), comparing the omission-induced dip to a zero may not be appropriate. A better baseline might be using the signals prior to the cue.

• Could the authors comment on how they came up with the 4-5.3s window to observe the AUC in Fig 3H?

These have all been addressed.

Minor

• When discussing the understudied role of dopamine in brain regions other than the striatum in the Introduction, it might be helpful to cite this article: https://elifesciences.org/articles/81980 where the authors characterize dopamine in the bed nucleus of stria terminalis in associative behaviors and reward prediction error.

• In Discussion, it might be better to refrain from describing the results as 'measuring dopamine release' in the LH. Since there was no direct detection of dopamine release, rather dopamine binding to the dLight receptors, referring to the detection as dopamine signaling/binding/transients is a better alternative.

• In Discussion, without measuring tonic dopamine release, it is difficult to say that there was a tonic dopamine release in the LH prior to negative RPE. In addition, I wouldn't describe the negative RPE as silencing of dopamine neurons projecting to the LH since this was not directly measured and it is hard to say for sure if the dip in dopamine is caused by silencing of the neurons. There certainly seems to be a reduction in extrasynaptic dopamine signaling in LH, however what occurs upstream is unknown.

• Typo at multiple places: 'Tekey's multiple comparison test'.

These have been addressed.

---

## [Author Response]

The following is the authors’ response to the original reviews.

**Reviewer #1 (Public Review):**
Summary:Mice can learn to associate sensory cues (sound and light) with a reward or activation of dopamine neurons in the ventral tegmental area (VTA), and then anticipate the reward from the sensory cue only. Using this paradigm, Harada et al. showed that after learning, the cue is able to induce dopamine release in the projection targets of the VTA, namely the nucleus accumbens and lateral hypothalamus (LH). Within the LH, dopamine release from VTA neurons (either by presentation of the cue or direct optical stimulation of VTA neurons) activates orexin neurons, measured as an increase in intracellular calcium levels.Strengths:This study utilized genetically encoded optical tools to selectively stimulate dopamine neurons and to monitor dopamine release in target brain areas and the calcium response of orexin neurons. This allowed a direct assessment of the relationship between the behavioral response of the animals, the release of a key neurotransmitter in select brain areas, and its effect on target cells, with a precision previously not possible. The results shed light on the mechanism underlying reward-related learning and expectation.Weaknesses:The Ca increase in orexin neurons in response to optical stimulation of VTA DA neurons is convincing. However, there is an accumulated body of literature indicating that dopamine inhibits orexin neurons through D2 receptors, particularly at high concentrations both directly and indirectly (PMID 15634779, 16611835, 26036709, 30462527; but note that synaptic effects at low conc are excitatory - PMID 30462527, 26036709). There should be a clear acknowledgment of these previous studies and a discussion directly addressing the discrepancy. Furthermore, there are in-vivo studies that investigated the role of dopamine in the LH involving orexin neurons in different behavioral contexts (e.g. PMID 24236888). The statement found in the introduction "whether and how dopamine release modulates orexin neuronal activity has not been investigated vigorously" (3rd para of Introduction) is an understatement of these previous reports.

We thank the Reviewer for pointing out that we missed several important citations. We added the references mentioned and the discrepancy of concern is addressed in the discussion section

Along these lines, previous reports of concentration-dependent bidirectional dopaminergic modulation of orexin neurons suggest that high and low levels of DA would affect orexin neurons differently. Is there any way to estimate the local concentration of DA released by the laser stimulation protocol used in this study?Could there be a dose dependency in the Intensity of laser stimulation and orexin neuron response?

We agree that this is an interesting point. However, one limitation of our study, and of intensity-based genetically-encoded sensors in general, is that the estimation of the concentration is technically difficult. The sensor effectively reports changes in extra-synaptic levels of neurotransmitters, but to get the absolute value other modalities would be needed such as fast scan voltammetry. This limitation is now included in the discussion section.

The transient dip in DA signal during omission sessions in Fig2C (approx 1% decrease from baseline) is similar in amplitude compared to the decrease seen in non-laser trails shown in Fig 1C right panel (although the time course of the latter is unknown as the data is truncated). The authors should clarify whether those dips are a direct effect of the cue itself or indeed reward prediction error.

Thanks for raising this important point. Indeed, there is a dip of the signal during non-stimulation trials. At day 1, the delivery of the cue triggered a dip and at day 10, there was a slight increase of the signal and followed by the dip. The data is difficult to interpret but our hypothesis is that two components trigger this dip of the signal. One is the aversiveness of the cue. Because a relatively loud sound (90dB) was used for the cue, it would not be surprising if the auditory cue was slightly aversive to the experimental animals. It has been shown that aversive stimuli induce a dip of dopamine in the NAc, although it is specific to NAc subregions. The second component is reward prediction error. Although the non-laser paired cue never triggered the laser stimulation, it is similar to the laser paired one. In a way both are composed of loud tone and same color of the visual cue (spatially different). We think it is possible that reward-related neuronal circuit was slightly activated by the non-laser paired cue. In line with this interpretation, a small increase of the signal was observed at day 10 but not day 1. If our hypothesis is true, since this signal was induced by two components, further analysis is unfortunately difficult.

There seem to be orexin-negative-GCaMP6 positive cells (Fig. 4B), suggesting that not all cells were phenotypically orexin+ at the time of imaging.The proportion of GCaMP6 cells that were ORX+ or negative and whether they responded differently to the stimuli should be indicated.

While we acknowledge the observation of orexin-negative-GCaMP6 positive cells in Figure 4B, it's important to note that this phenomenon is consistent with the characteristics of the hOX-GCaMP virus used in prior experiments. The virus has undergone thorough characterization, and it has been reported to exhibit over 90% specificity, as demonstrated in prior work conducted in the laboratory of one of our contributing authors (PMID: 27546579). To address the concern raised by the reviewer, we have included Supplemental Figure 4 confirming that all mice consistently exhibited qualitatively similar hOX-GCaMP transients upon dopaminergic terminal stimulation. This additional evidence supports the reliability and specificity of our experimental approach.

Laser stimulation of DA neurons at the level of cell bodies (in VTA) induces an increase in DA release within the LH (Fig. 3C, D), however, there is no corresponding Ca signal in orexin neurons (Fig.4C).

We realized that the figures were not clear and we understood that the reviewer did not see any corresponding Ca signal, but this description is not true. We now added Supplemental Figure 3 to show that there is Ca signal at day 1 already.

In contrast, stimulating DA terminals within the LH induces a robust, long-lasting Ca signal (> 30s) in orexin neurons (Fig. 5). The initial peak is blocked by raclopride but the majority of Ca signal is insensitive to DA antagonists (please add a positive control or cite references indicating that the dose of antagonists used was sufficient; also the timing of antagonist administration should be indicated).

This is now included in the discussion section. Also, the timing and dose of the antagonist is now described in the method section.

Taken together, these results seem to suggest that DA does not directly increase Ca signal in orexin neurons. What could be mediating the remaining component?

This point has been included in the discussion section.

Similarly, there is an elevation of Ca signal in orexin neurons that remains significantly higher after the cue/laser stimulation (Fig. 4F). It appears that it is this sustained component that is missing in omission trials.This can be analyzed further.

It is true that there is a sustained component in stimulation trials, that is missing in omission trials. Most likely that is evoked by the stimulation of dopamine neurons. We argue that this component is isolated in Fig 5 and analyzed as much as we can.

Mice of both sexes were used in this study; it would be interesting to know whether sex differences were observed or not.

We agree that this is an important point. However, our sample number is not high enough to make a meaningful comparison between male and female.

**Reviewer #2 (Public Review):**
Summary:This is an interesting and well-written study assessing the role of dopaminergic inputs from the VTA on orexin cell responses in an opto-pavlovian conditioning task. These data are consistent with a possible role of this system in reward expectation and are surprisingly one of the first demonstrations of a role for dopamine in this phenomenon.Strengths:The study has used an interesting opto-Pavlovian approach combined with fibre photometry.Weaknesses:It is unclear what n size was used or analysed, particularly for AUC measures e.g. Figures 1 D/E and 3 G.The number of trials reflected and the animal numbers need clarification.

The sample size is indicated in the legend section.

The study focused on opto-stim omissions - this work would be significantly strengthened by a comparison to a real-world examination where animals are trained for a radiation reward (food pellet).

We agree that this would be an important experiment. This experiment is partially done in one of the contributing authors laboratories (doi.org/10.1101/2022.04.13.488195) and would be one of our follow up study.

Have the authors considered the role of orexin in the opposing situation i.e. a surprise addition of reward?

That would be an interesting experiment. To do that, natural reward, not optical stimulation, should be used as a reinforcer. This could be part of our follow up study.

Similarly, there remains some conjecture regarding the role of these systems in reward and aversion - have the authors considered aversive learning paradigms - fear, or fear extinction - to further explore the roles of this system? There are some (important) discussions about the possible role of orexin in negative reinforcement. Further studies to address this could be warranted.

It is true that dopamine also plays a significant role in aversive learning. Therefore, this would be an interesting experiment. The discussion section now includes this point.

I think some further discussion of the work by Lineman concerning the interesting bidirectional actions of d1/d2 r signalling on glutamatergic transmission onto orexin neurons is worthwhile. While this work is currently cited, the nuance and perhaps relevance to d1 and d2 signalling could be contextualised a little more (https://doi.org/10.1152/ajpregu.00150.2018).

Thanks for the suggestion. The discussion has been expanded.

**Reviewer #3 (Public Review):**
Summary:Harada and colleagues describe an interesting set of experiments characterizing the relationship between dopamine cell activity in the ventral tegmental area (VTA) and orexin neuron activity in the lateral hypothalamus (LH). All experiments are conducted in the context of an opto-Pavlovian learning task, in which a cue predicts optogenetic stimulation of VTA dopamine neurons. With training, cues that predict DA stimulation come to elicit dopamine release in LH (a similar effect is seen in accumbens). After training, omission trials (cue followed by no laser) result in a dip (inhibition) of dopamine release in LH, characteristic of reward prediction error observed in the striatum. Across cue training, the activity pattern of orexin neurons in LH mirrors that of LH DA levels. However, unlike the DA signal, orexin neurons do not exhibit a decrease in activity in omission trials. Systemic blockade of D2 but not D1 receptors blocked DA release in LH following VTA DA cell stimulation.Strengths:Although much work has been dedicated to examining projections from orexin cells to VTA, less has been done to characterize reciprocal projections and their function. In this way, this paper is a very important addition to the literature. The experiments are technically sound (with some limitations, below) and utilize sophisticated approaches, the manuscript is nicely written, and the conclusions are mostly reasonable based on the data collected.Weaknesses:I believe the impact of the paper could be enhanced by considering and/or addressing the following:Major:I encourage the authors to discuss in the Introduction previous work on DA regulation of orexin neurons. In particular, the authors cite, but do not describe in any detail, the very relevant Linehan paper (2019; Am J Physiol Regul) which shows that DA differentially alters excitatory/inhibitory input onto orexin neurons and that these actions are reversed by D1 vs D2 receptor antagonists. Another paper (Bubser, 2005, EJN) showed that dopamine agonists increase the activity of orexin neurons and that these effects are blocked by D1/D2 antagonists. The current findings should be discussed in the context of these (and any other relevant) papers in the Discussion, too.

Thanks for the valuable suggestion. This point has been integrated and the introduction and discussion sections have been revised carefully.

In the Discussion, the authors provide two (plausible) explanations for why they did not observe a dip in the calcium signal of orexin neurons during omission trials. Is it not possible that these cells do not encode for this type of RPE?

We completely agree that it is possible. Now our current hypothesis is that dopamine in the LH encodes RPE and that information is transmitted to orexin neurons. Orexin neurons integrate other information and encode something else, we call it ‘multiplexed cognitive information’. It is still open question what this means exactly. This point is now mentioned in the discussion section.

Related to the above - I am curious about the authors' thoughts on why there is such redundancy in the system. i.e. why is dopamine doing the same thing in NAC and LH in the context of cue-reward learning?

Thank you for the question. This is an important point, indeed. Our current hypothesis is described in the discussion section.

’Our data indicate that dopamine in both the NAc and LH encodes reward prediction error (RPE). One open question is the existence of such a redundant mechanism. We hypothesize that dopamine in the LH boosts dopamine release via a positive feedback loop between the orexin and dopamine systems. It has already been established that some orexin neurons project to dopaminergic neurons in the VTA, positively modulating firing. On the other hand, our data indicate that dopamine in the LH stimulates orexinergic neurons. These collective findings suggest that when either the orexin or dopamine system is activated, the other system is also activated consequently. Although the current findings align with this idea, the hypothesis should be carefully challenged and scrutinized.’

The data, as they stand, are largely correlative and do not indicate that DA recruitment of orexin neurons is necessary for learning to occur. It would be compelling if blocking the orexin cell recruitment affected some behavioral outcomes of learning. Similarly - does raclopride treatment across training prevent learning?

We appreciate the insightful comment. It is indeed a limitation of our study that we lack behavioral data. However, given the extensive previous research on the crucial role of orexin in motivated behavior, we argue that establishing dopaminergic regulation of the orexin system itself is a valuable contribution. This perspective is thoroughly discussed in the dedicated section of our paper. It's important to note that the injection of D2 antagonists, including raclopride, is known to induce significant sedation. Due to this sedative effect, combining behavioral experiments with these drugs poses considerable challenges.

Only single doses of SCH23390 and raclopride were used. How were these selected? It would be nice to use more of a dose range to show that (1) and effect of D1R blockade was not missed, and (2) that the reduction in orexin signal with raclopride was dose-dependent.

The rationale of the dose has been added to the discussion session. It is reported that these doses block dopamine receptors. We agree that it would be nice to have a dose-response curve, we are reluctant to increase the doses to avoid adverse effect to the experimental animals. The doses we used effectively induced hypo-locomotion, although data is not shown.

Fig 1C, could the effect the authors observed be due to movement?

We argue this is unlikely. We recorded two channels one for the control and the other one for the signal. The motion-related artifact is corrected based on the control channel. One example trace around the laser stimulation is shown below. Please note that a typical motion-related artifact is a fast dip of the signal, normally observed in both 405 and 465 nm channels.

Relatedly, what was the behavior like when the cue was on? Did mice orient/approach the cue?

Although it has been reported that rats approach the cue (PMID: 30038277) in a similar task, it was not obvious in our case. It could be because we used both visual and auditory cues. Mice showed a general increase of locomotion during the cue and the stimulation but the direction was not clear to the experimenter.

Also, when does the learning about the cue occur? Does it take all 10 days of learning or does this learning/cue-induced increase in dopamine signaling occur in less than 10 days?

It is hard to say when the learning occurs. When we look at the learning curve of Figures 1,3 and 4, it seems the response to the cue plateaus at day 5 but since we don’t have behavioral data, the assessment is relayed only on the neuronal signal.

Also related to the above, could the observed dopamine signal be a result of just the laser turning on? It would seem important to include mice with a control sensor.

We recorded two channels, 405 nm and 465 nm wavelength. 405 nm signal did not show increase of the signal while 465 nm signal did. The example trace is shown. Besides, the sensor has been characterized by the corresponding author already so we argue that this is unlikely.

Fig 1E, the effect seems to be driven by one mouse which looks like it could be a statistical outlier. The inclusion of additional animals would make these data more compelling.

We agree that adding more mice would make data more compelling. However, considering the fact that dopamine in the accumbens has been investigated vigorously and our data is in line with the prior studies, we argue that we have enough data to claim our conclusion.

For Fig 1C, 3D, 3F, and 4D, could the authors please show the traces for the entire length of laser onset? It would be helpful to see both the rise and the fall of dopamine signals.

For Fig 1C, one panel has been added. For fig 3, 4, supplemental figure was created to show the signal around laser stimulation.

Fig 2C, could the authors comment on how they compared the AUC to baseline? Was this comparison against zero? Because of natural hills and troughs during signals prior to cue (which may not equate to a zero), comparing the omission-induced dip to a zero may not be appropriate. A better baseline might be using the signals prior to the cue.

The signal immediately before the cue onset was considered as a baseline, and baseline was subtracted. This means zero and baseline would be the same in our way of analysis.

Could the authors comment on how they came up with the 4-5.3s window to observe the AUC in Fig 3H?

Since the kinetic of dopamine in the NAc and LH is different, different time windows have been used to observed a dip of dopamine. The analysis of the kinetics has been added.

**Recommendations for the authors:**

**Reviewer #1 (Recommendations For The Authors):**
Specific feedback to the authorsSample size for each experiment/group could not be found.

The sample size is now included in the legends.

In most figures, the timing of onset for the cue and laser stimulation is unclear. This makes the data interpretation difficult. They should be labeled as in Fig. 3C, for example.

Panels have been updated to address this point.

Please provide the rationale for selecting the time range for the measurement of AUC for different experiments (e.g. Fig. 2C, 3H, 4A, 5F).

The kinetics of dopamine in NAc and LH are different. This is now shown in the new Supplemental Figure 2. Based on this difference, the different window was chosen.

Fig. 1E, 3G right, 4E right: statistical analysis should use two-way repeated measures ANOVA rather than one-way ANOVA. Fig 1D, 3G left and 4E left panels can also be analyzed by two-way repeated measures ANOVA.

We realized that those panels were redundant. Some panels have been removed and the analysis has been conducted according to this point.

Minor comments:Fig. 2C can also show non-omission trials as a comparison.

The panel has been updated.

The term "laser cue" is confusing, as the cue itself does not involve a laser.

’Laser-paired cue’ is used instead.

Color contrast can be improved for some figures, including Fig. 2C right, Fig. 3H right, and green and blue fluorescent fonts.

The panels have been updated.

Figure legends: Tukey's test, rather than Tekey's test.

This has been fixed.

There are some long-winded sentences that are hard to follow.

Edited.

p.2, line 11 from bottom: should read ...the VTA evokes the release of dopamine.

Edited

p.3, line 9: remove e from release.

This has been addressed.

**Reviewer #3 (Recommendations For The Authors):**
Minor:When discussing the understudied role of dopamine in brain regions other than the striatum in the Introduction, it might be helpful to cite this article: https://elifesciences.org/articles/81980 where the authors characterize dopamine in the bed nucleus of stria terminalis in associative behaviors and reward prediction error.

The discussion session has been updated accordingly.

In the Discussion, it might be better to refrain from describing the results as 'measuring dopamine release' in the LH. Since there was no direct detection of dopamine release, rather a dopamine binding to the dLight receptors, referring to the detection as dopamine signaling/binding/transients is a better alternative.

This point has been addressed.

In the Discussion, without measuring tonic dopamine release, it is difficult to say that there was a tonic dopamine release in the LH prior to negative RPE. In addition, I wouldn't describe the negative RPE as silencing of dopamine neurons projecting to the LH since this was not directly measured and it is hard to say for sure if the dip in dopamine is caused by silencing of the neurons. There certainly seems to be a reduction in extra-synaptic dopamine signaling in LH, however, what occurs upstream is unknown.

We respectfully disagree with this point. In our opinion, the dopamine transient is more important than the firing of dopamine neurons because what matters for downstream neurons is dopamine concentration. For example, administration of cocaine increases the dopamine concentration extra-synaptically via blockade of DAT, while the firing of dopamine neurons go down via activation of D2 receptors expressed in dopamine neurons. Administration of cocaine is not known to induce negative RPE.

Typo at multiple places: 'Tekey's multiple comparison test'.

This has been fixed.